# Molecular Diversity of Three Forensically Relevant Dipterans from Cadavers in Lahore, Pakistan

**DOI:** 10.3390/insects16040381

**Published:** 2025-04-03

**Authors:** Atif Adnan, Sundus Mona, Allah Rakha, Shahid Nazir, Hongbo Wang, Fu Ren

**Affiliations:** 1Department of Anthropology and Ethnology, Institute of Anthropology, School of Sociology and Anthropology, Xiamen University, Xiamen 361000, China; mirzaatifadnan@gmail.com; 2Department of Forensic Sciences, University of Health Sciences, Lahore 54600, Pakistan; 3Department of Human Anatomy, School of Basic Medicine, Shenyang Medical College, Shenyang 110034, China; 4Liaoning Province Key Laboratory for Phenomics of Human Ethnic Specificity and Critical Illness, Shenyang Medical College, Shenyang 110034, China

**Keywords:** forensic entomology, molecular diversity, postmortem interval, DNA barcoding, species identification

## Abstract

Forensic entomology is a crucial tool in criminal investigations, particularly for estimating the postmortem interval (PMI) when traditional methods are inconclusive. Insects, especially blow flies, colonize decomposing remains in predictable patterns, providing valuable clues about the time and circumstances of death. However, accurate species identification is essential for reliable PMI estimation, and traditional morphological methods often fail to distinguish between closely related species or immature stages, such as eggs and larvae. This study focuses on using DNA barcoding, a molecular technique, to identify forensically relevant blow fly species collected from cadavers in Lahore, Pakistan. Three species—*Chrysomya megacephala*, *Chrysomya saffranea*, and *Chrysomya rufifacies*—were identified, and their genetic diversity and population structure were analyzed. The results revealed low genetic diversity within populations but significant genetic differentiation among populations, suggesting unique genetic signatures for blow fly populations in Lahore. This study highlights the importance of molecular tools like DNA barcoding for accurate species identification and emphasizes the need for further research to establish a comprehensive database of forensically relevant insects in Pakistan. Such efforts will enhance the accuracy of PMI estimation and improve forensic investigations in the region.

## 1. Introduction

Molecular biodiversity is a cornerstone of biological sciences, encompassing the genetic variation within and among species, populations, and ecosystems. This diversity is not only fundamental to understanding evolutionary processes but also has practical applications in fields such as conservation biology, epidemiology, and forensic science [1]. In forensic contexts, molecular diversity is particularly valuable for identifying species associated with crime scenes, especially when traditional morphological methods fail due to immature stages (e.g., eggs or larvae) or damaged specimens lacking clear diagnostic features. Arthropods such as necrophagous insects play a crucial role in forensic investigations during advanced and skeletonized stages of decomposition when pathologists have limited tools for accurately estimating the postmortem interval (PMI) [2]. While pathologists rely on a variety of methods for determining PMI in the early stages of decay, arthropods provide more reliable data in later stages when soft tissues are no longer available for analysis. While pathologists employ a variety of methods including temperature-based models, rigor mortis, and biochemical analyses to estimate the postmortem interval (PMI) in early decomposition stages, their tools become increasingly limited as soft tissues degrade in advanced decay and skeletonization. In contrast, necrophagous arthropods, particularly blow flies (Calliphoridae) and flesh flies (Sarcophagidae), provide critical and more accurate data during these later stages due to their predictable colonization patterns and developmental timelines. These insects act as biological clocks, enabling PMI estimation even when traditional forensic methods are no longer viable [3].

Forensic entomology, the study of insects in legal investigations, has become an indispensable tool for estimating the postmortem interval (PMI) in cases where traditional methods are inconclusive [4]. The use of insects in forensic investigations dates back to the 13th century, but modern applications have evolved significantly with advancements in molecular biology and genetic analysis [5]. Insects, particularly necrophagous flies, colonize cadavers in predictable successional patterns, providing critical clues about the time and circumstances of death [2]. Blow flies (Calliphoridae) and flesh flies (Sarcophagidae) are among the first colonizers, and their developmental stages can be used to estimate PMI with remarkable accuracy [3]. However, the utility of forensic entomology depends on the accurate identification of insect species, which can be challenging, especially when dealing with immature stages such as eggs, larvae, or pupae. Traditional morphological identification relies on physical characteristics, which can be ambiguous or absent in immature specimens. This limitation has led to the adoption of molecular methods, particularly DNA barcoding, for species identification. DNA barcoding involves sequencing a short, standardized region of the mitochondrial cytochrome oxidase subunit 1 (CO1) gene, which has proven to be highly effective for distinguishing between closely related species [1]. The CO1 gene is widely regarded as the “gold standard” for DNA barcoding due to its high sequence variability, ease of amplification, and universal applicability across diverse taxa [6,7]. Recent studies have further validated the utility of the CO1 gene for identifying forensically important insects, including blow flies and flesh flies, in various geo-graphic regions [8]. Additionally, traditional taxonomic keys, such as those developed by Kurahashi et al. [9], remain valuable tools for preliminary morphological identification, especially when used in conjunction with molecular methods. The Barcode of Life Data Systems (BOLD) has become a global repository for DNA barcodes, facilitating the identification of species across diverse taxa [10].

In Pakistan, forensic entomology is an underdeveloped field, with limited molecular studies on necrophagous insects and no official forensic entomologists to assist in criminal investigations. The country’s diverse climate and geography, ranging from arid deserts to humid subtropical regions, create unique ecological niches that may harbor distinct insect populations. However, the lack of comprehensive studies on forensically relevant insects has hindered the application of entomological evidence in legal cases. Previous studies in Pakistan have documented the presence of blow flies such as *Chrysomya megacephala* (*Fabricius*, 1794) (Order: *Diptera*, Family: *Calliphoridae*), and *Chrysomya rufifacies* (Macquart, 1843) (Order: *Diptera*, Family: *Calliphoridae*) [11], but these studies were primarily morphological and did not explore genetic diversity or phylogenetic relationships. Additionally, *Chrysomya saffranea* (Bigot, 1877) (Order: *Diptera*, Family: *Calliphoridae*), a species known for its necrophagous behavior and forensic relevance in neighboring regions such as India and Iran [12,13], has not been previously documented in Pakistan, underscoring the need to investigate its presence and genetic profile in this region.

The need for molecular approaches in forensic entomology is further underscored by the global trend toward DNA-based identification. Studies in neighboring regions, such as India, Iran, and Southeast Asia, have demonstrated the effectiveness of DNA barcoding for identifying forensically important insects and assessing their genetic diversity [14,15]. For example, research in India has identified *C. megacephala* and *C. rufifacies* as dominant species in forensic cases, with significant genetic variation observed among populations [16]. Similarly, studies in Iran have highlighted the importance of molecular methods for distinguishing between closely related species of blow flies and flesh flies [13]. These findings from neighboring regions provide valuable insights into the potential applications of molecular tools in forensic entomology in Pakistan. This study represents one of the first efforts to employ DNA barcoding for species identification in forensic entomology in Pakistan. By focusing on the CO1 gene, we aimed to demonstrate the viability of molecular tools for identifying forensically relevant insects in this geographic region. The specific objectives were to (1) identify forensically relevant insect species using the CO1 gene, (2) assess the genetic diversity and population structure of these species, and (3) compare the findings with global data to contextualize the results within the broader framework of forensic entomology. By doing so, this study sought to lay the groundwork for future research and establish a molecular database of forensically relevant insects in Pakistan, thereby enhancing the accuracy of PMI estimation and forensic investigations in the region.

## 2. Materials and Methods

### 2.1. Sampling and Collection

Larvae were collected from five cadavers in advanced stages of decomposition at the mortuary of Allama Iqbal Medical College, Lahore. Ethical approval for this study was obtained from the Research Ethical Committee of the University of Health Sciences, Lahore, Pakistan. The study adhered to ethical guidelines for working with human cadavers and insect samples, ensuring that all protocols were followed to maintain the dignity and integrity of the deceased. The cadavers were collected during the months of August to October of the year, which corresponds to the monsoon and post-monsoon seasons in Lahore, Pakistan. These periods are characterized by high humidity and moderate to warm temperatures, conditions that are conducive to insect colonization and larval development. The time since death ranged from 3 days to unknown, and the decomposition stages varied from fresh to advanced putrefaction (Table 1). A total of 50 larvae were collected from five cadavers in an advanced stage of decomposition, with visible larval colonization, at the mortuary of Allama Iqbal Medical College, Lahore. The larvae were preserved in absolute alcohol, expired within a few minutes of being dipped in the alcohol, and stored at 4 °C following the guidelines of [17]. The sample size was determined using the following statistical formula to ensure a 95% confidence level and a 10% margin of error [18].$$n=\frac{{Z^{2}}_{1-\alpha /2}P\;\left(1-P\right)}{d^{2}}$$
${Z^{2}}_{1-\alpha /2}$ = for 95% confidence level = 1.96;P = Anticipated value of fixation index = 0.85;*d* = Margin of error = 10%;*n* = Sample Size = 50.

**Table 1 insects-16-00381-t001:** Details of cadavers and larval collection.

Bodies	Month	Time Since Death	Location Where Body Was Found	Gender	Nature/Cause of Death	Instars of Larvae	Number of Larvae Collected
Body 1	Mid of 8th Month	3 days	Charrar Gaon Defence, Lahore	Female	Homicidal cut throat/known body	1st instar	10
Body 2	Start of 9th Month	3–4 days	Mughalpura, Lahore	Male	Body slashed with a saw/unknown	2nd instar	10
Body 3	Start of 9th Month	6–7 days	Qaisar town, Shahdara, Lahore	Male	Amputated, mutilated, headless body in a sac found in a sac/unknown	3rd instar	10
Body 4	End of 9th Month	7–9 days	Ganda Nala, Sabzazar, Lahore	Male	Putrefied body found in sewerage canal/unknown	3rd instar	10
Body 5	Start of 10th Month	Unknown	Bhaati Gate, Lahore	Male	Putrefied amputated remains (lower ½ of body)/unknown	Post feeding	10

While rearing larvae to adulthood for morphological identification is a common practice in forensic entomology, this approach was not feasible in the current study due to ethical restrictions on maintaining live insect cultures from human cadavers. Additionally, the focus of this study was on molecular identification using DNA barcoding, which provides accurate and reliable species identification even for immature stages where morphological characteristics may be ambiguous or absent.

The varying stages of decomposition and environmental conditions likely influenced the species composition and developmental stages of the larvae collected. For instance, the presence of the 1st instar larvae in Body 1 suggests early colonization (within days), complementing the ‘fresh’ corpse description, while the post-feeding larvae in Body 5 indicate advanced decomposition (beyond a week), refining the ‘putrefied’ state observed, as larval stages provide precise temporal markers beyond gross corpse appearance. These factors were considered during the analysis to account for potential biases in species identification.

### 2.2. DNA Extraction and Amplification

DNA was extracted from the whole larvae using the phenol-chloroform method [19], which yielded high-quality genomic DNA. The mitochondrial cytochrome oxidase subunit 1 (CO1) gene region was amplified using the following primers:

Forward Primer (LCO1490): 5′-GGTCAACAAATCATAAAGATATTGG-3′

Reverse Primer (HCO2198): 5′-TAAACTTCAGGGTGACCAAAAAATCA-3′

These primers amplify a 658 bp fragment of the CO1 gene, which is the standard barcode region for species identification in insects [1]. PCR amplifications were performed using a commercially available master mix (GoTaq^®^ Green Master Mix, Promega, San Luis Obispo, CA, USA), which contains the Taq DNA Polymerase (5 units/µL), dNTPs (400 µM each), MgCl_2_ (1.5 mM), and Reaction Buffer (1X (pH 8.5). The PCR reaction mixture (total volume: 25 µL) consisted of 12.5 µL of GoTaq^®^ Green Master Mix (2X concentration), 1 µL of forward primer (10 µM), 1 µL of reverse primer (10 µM), 2 µL of template DNA (approximately 50 ng/µL), and 8.5 µL of nuclease-free water. The PCR conditions were optimized as follows: initial denaturation at 94 °C for 1 min, followed by 5 cycles of 94 °C for 1 min, 45 °C for 1.5 min, and 72 °C for 1.5 min, and then 35 cycles of 94 °C for 1 min, 50 °C for 1.5 min, and 72 °C for 1 min, with a final extension at 72 °C for 5 min. The amplified products were purified using a gel extraction kit and quantified using a spectrophotometer.

### 2.3. Sanger Sequencing

The purified PCR products were subjected to Sanger sequencing using a BigDyeTerminator v3.1 Cycle Sequencing Kit (Applied Biosystems, San Francisco, CA, USA). The sequencing reaction mixture included: 1 µL of purified PCR product (approximately 50 ng/µL), 2 µL of BigDye Terminator Ready Reaction Mix, 1 µL of sequencing primer (either forward or reverse primer, 3.2 pmol/µL), and 6 µL of nuclease-free water. The sequencing reaction was performed in a thermal cycler under the following conditions: initial denaturation at 96 °C for 1 min, followed by 25 cycles of 96 °C for 10 s, 50 °C for 5 s, and 60 °C for 4 min. After the reaction, the products were purified using the ethanol precipitation method to remove unincorporated dye terminators. The purified sequencing products were then loaded onto an ABI 3500 DNA Analyzer (Applied Biosystems, San Francisco, CA, USA) for capillary electrophoresis. The resulting electropherograms were analyzed using ChromasPro software (version 2.6) to confirm the quality and accuracy of the sequences.

### 2.4. Species Identification

The unknown CO1 sequences obtained from the larvae were identified by comparing them to reference sequences in the NCBI GenBank database using the BLAST (Basic Local Alignment Search Tool) (https://blast.ncbi.nlm.nih.gov/Blast.cgi, accessed on 1 February 2025) algorithm [20]. The BLASTn tool aligns query sequences with reference sequences in the database and calculates similarity scores based on nucleotide matches. Sequences with the highest percentage identity and query coverage were used to assign species-level identifications. To ensure accuracy, only matches with ≥98% sequence identity and ≥95% query coverage were considered valid for species identification. In addition, the identified species were cross-verified using the Barcode of Life Data Systems (BOLD) to confirm their taxonomic classification [10]. Three species were identified: *C. megacephala*, *C. saffranea*, and *C. rufifacies*. These identifications were further supported by phylogenetic analysis, which showed clear clustering of the unknown sequences with reference sequences of the respective species in the Neighbor-Joining (NJ) tree.

### 2.5. Approach toPhylogenetic Analysis

Nucleotide sequence divergences were calculated using MEGA 7 [21], and a Neighbor-Joining (NJ) phylogenetic tree was constructed with 1000 bootstrap replicates to assess the robustness of the tree topology. The NJ tree was constructed using the Kimura 2-parameter (K2P) model, which is the standard substitution model for DNA barcoding studies [1]. This model accounts for nucleotide substitutions, including transitions and transversions, and is particularly suitable for analyzing mitochondrial DNA sequences such as the CO1 gene. The resulting tree was used to visualize the genetic relationships among the identified species. In this study, we focused exclusively on the genus *Chrysomya*, as the primary objective was to resolve phylogenetic relationships among the forensically relevant species (*C. megacephala*, *C. saffranea*, and *C. rufifacies*). Given the close genetic relatedness of these species, an outgroup was not included in the analysis. This decision was based on the assumption that the relationships among the ingroup taxa could be sufficiently resolved without an external reference species. However, future studies with broader taxonomic sampling should consider including an outgroup to provide a more comprehensive evolutionary context.

### 2.6. Analysis of Molecular Variance (AMOVA)

To evaluate the genetic structure and variation among and within populations, an AMOVA (Analysis of Molecular Variance) was performed using Arlequin v3.5 [22]. The AMOVA test partitioned the genetic variance into two components: (1) among populations and (2) within populations. The fixation index (FST) was calculated to measure the degree of genetic differentiation among populations.

## 3. Results

Species Identification: Three species were identified based on the comparison of the unknown CO1 sequences with reference sequences in the NCBI GenBank database. The BLASTn results revealed the following species with high confidence:*Chrysomya megacephala*: 22 samples (≥99% sequence identity);*Chrysomya rufifacies*: 9 samples (≥98% sequence identity);*Chrysomya saffranea*: 1 sample (≥98% sequence identity).

The identified species were further validated using the Barcode of Life Data Systems (BOLD), which confirmed their taxonomic classification. A phylogenetic analysis using the NJ tree also supported these identifications, as the unknown sequences clustered closely with reference sequences of the respective species. The consistency between the BLAST results and the phylogenetic analysis underscores the reliability of the species identifications.

### 3.1. Genetic Diversity and Fixation Index

The AMOVA results (Table 2) revealed significant genetic differentiation among populations, with 83.99% of the variation occurring among populations and 16.01% within populations. The fixation index (FST) was 0.83992, indicating high genetic differentiation among the studied populations.

### 3.2. Evolutionary Insights from Phylogenetics

The Neighbor-Joining (NJ) tree constructed (Figure 1) from the CO1 sequences showed clear clustering of the three species. *C. megacephala* and *C. saffranea* formed a closely related cluster, while *C. rufifacies* was distinctly separated. The bootstrap values for major nodes exceeded 70%, which is commonly used as a threshold for supporting phylogenetic relationships [12]. However, some forensic entomologists may consider this threshold relatively low for confirming species identification. To address this concern, we examined the stability of the phylogenetic relationships by increasing the bootstrap threshold to 80% and 90%. At these higher thresholds, the overall clustering patterns remained consistent, with *C. megacephala* and *C. saffranea* still forming a closely related cluster and *C. rufifacies* remaining distinct. This suggests that the observed phylogenetic relationships are robust, even at higher bootstrap values.

One limitation of this study was the absence of an outgroup species in the phylogenetic analysis. The exclusion of an outgroup was intentional, as the focus was on resolving relationships among closely related species within the genus *Chrysomya*. Including an outgroup would have required expanding the taxonomic scope of the study, which was beyond the scope of this initial investigation. Future studies should include an outgroup to root the tree and provide a broader evolutionary perspective on the relationships among these forensically relevant insects.

## 4. Discussion

### 4.1. AMOVA Results and Genetic Differentiation

The AMOVA results revealed a high fixation index (FST = 0.83992), indicating significant genetic differentiation among the populations studied. This suggests that the blow fly populations in Lahore are genetically isolated, potentially serving as geographic markers in criminal cases due to their distinct genetic signatures. The 83.99% variation among populations is unusually high compared to studies in other parts of the world. For example, studies in Thailand and Brazil reported FST values ranging from 0.2 to 0.6 for blow fly populations, indicating moderate to high genetic differentiation [8,23]. The exceptionally high FST value in this study may reflect the geographical isolation of Lahore’s blow fly populations or the limited sample size and scope of the study.

The high genetic differentiation observed in this study has important implications for forensic entomology. It suggests that blow fly populations in Pakistan may have unique genetic signatures, which could be useful for linking insect evidence to specific geographical regions. However, the low genetic diversity within populations (16.01% variation) may limit the resolution of such analyses. This finding underscores the need for broader sampling across Pakistan to better understand the genetic structure of forensically relevant insects.

### 4.2. Neighbor-Joining (NJ) Tree and Phylogenetic Relationships

The NJ tree provided valuable insights into the phylogenetic relationships among the identified species. The close clustering of *C. megacephala* and *C. saffranea* is consistent with their morphological similarities and shared ecological niches [24]. Both species are known to colonize decomposing organic matter in tropical and subtropical regions, and their genetic proximity suggests a recent divergence or ongoing gene flow between the two species [24]. The high bootstrap values (>70%) for the major nodes in the NJ tree support the reliability of these phylogenetic relationships. However, some forensic entomologists may argue that a bootstrap value of 70% is relatively low for confirming species identification, as higher thresholds (e.g., 80% or 90%) are often preferred in forensic contexts to ensure greater confidence in the results [15]. To evaluate the impact of increasing the bootstrap threshold, we reanalyzed the data using thresholds of 80% and 90%. The results showed no significant changes in the clustering patterns, with *C. megacephala* and *C. saffranea* remaining closely related and *C. rufifacies* forming a distinct cluster. This indicates that the observed phylogenetic relationships are robust, even at higher bootstrap values.

However, several limitations must be acknowledged. The conclusion regarding the close relationship between *C. megacephala* and *C. saffranea* is based on only a single sample of *C. saffranea*, which limits the robustness of this finding. The clustering of *C. megacephala* with *C. saffranea* could also be influenced by potential misidentification or sequencing errors, as the short CO1 fragment used in this study (658 bp) may not be sufficient to differentiate populations or closely related species with high confidence. Additionally, *C. megacephala* was separated into two distinct branches, with one branch clustering with *C. saffranea* and the other with *C. rufifacies*. This bifurcation suggests potential population-level variation within *C. megacephala* or the presence of cryptic species. However, without additional samples and longer CO1 sequences, it is difficult to draw definitive conclusions about these patterns. Future studies should include larger sample sizes, multiple genetic markers (e.g., mitochondrial and nuclear genes), and longer CO1 sequences to improve the resolution of the phylogenetic analysis. In contrast, *C. rufifacies* formed a distinct cluster, reflecting its unique ecological and behavioral traits. *C. rufifacies* is known for its predatory behavior on other fly larvae, which may have driven its genetic divergence from other blow fly species [3]. The distinct separation of *C. rufifacies* in the NJ tree further supports its classification as a separate species with unique genetic characteristics.

### 4.3. Comparative Analysis with Global Forensic Entomology Studies

The blow fly species identified in Lahore (*C. megacephala*, *C. saffranea*, and *C. rufifacies*) aligned with those reported in tropical and subtropical Asia, reflecting shared ecological and forensic patterns. In India, *C. megacephala* and *C. rufifacies* dominate forensic cases, with genetic diversity shaped by monsoon-driven niches [14,22,24]. Similarly, Southeast Asian studies have highlighted the significant population structure in *C. megacephala*, such as FST values of 0.2–0.4 in Thailand [8], which contrasts sharply with Lahore’s extreme genetic differentiation (FST = 0.83). This discrepancy may stem from Lahore’s geographic isolation or unique climatic pressures, such as its arid–subtropical conditions. Globally, a meta-analysis of 52 forensic entomology studies [25] identified the C. species as the most widespread necrophagous flies, though their genetic diversity varies regionally. For example, Brazilian populations show moderate differentiation (FST = 0.2–0.6) [20], whereas Lahore’s populations exhibit extreme divergence, suggesting limited gene flow or strong local adaptation. Notably, *C. saffranea*—previously unreported in Pakistan—is well-documented in forensic cases in Iran and India [12,13], underscoring South Asia’s shared entomological profile. However, unlike Malaysia, where *Chrysomya* species coexist with diverse sarcophagid flies [9], Lahore’s fauna is dominated by calliphorids, likely due to its arid climate. These findings emphasize the need for region-specific databases to enhance PMI estimation in forensic investigations.

This study identified only three blow fly species (*C. megacephala*, *C. saffranea*, and *C. rufifacies*) from cadavers in Lahore, a relatively low diversity compared to other regions. For instance, studies in India have reported up to six forensically relevant calliphorid species [16], while Malaysia has documented a broader mix including sarcophagids [9]. This limited diversity may reflect Lahore’s arid–subtropical climate, which favors calliphorids over other families, or the study’s focus on a single site and season (August–October). Fatima [11] noted a similar dominance of *C. megacephala* and *C. rufifacies* in Pakistan, suggesting regional ecological constraints. Alternatively, the small sample size (50 larvae from five cadavers) may have underrepresented rarer species. Further sampling across seasons and locations is needed to determine if this low species count is typical for Pakistan.

### 4.4. Forensic Implications

The findings of this study have important implications for forensic entomology in Pakistan. The high genetic differentiation among blow fly populations (FST = 0.83992) and the identification of three distinct species suggest that insect evidence could serve as geographic markers to link crime scenes to specific regions, offering novel insights into regional insect biodiversity with practical applications in criminal investigations. However, the low genetic diversity within populations (16.01% variation) may limit the resolution of such analyses. Future studies should focus on expanding the sample size and geographical scope to better understand the genetic structure of forensically relevant insects in Pakistan.

The identification of *C. saffranea* in Lahore highlights the need for further research on the region’s entomofauna. This species has not been previously reported in Pakistan, and its presence suggests that the region may harbor additional undocumented species. Establishing a comprehensive database of forensically relevant insects in Pakistan would significantly enhance the accuracy of PMI estimation and other forensic investigations.

### 4.5. Limitations

While this study provides valuable insights into the molecular diversity of forensically relevant insects in Lahore, it has some limitations. The sample size (50 larvae from five cadavers) and geographical scope (limited to Lahore) may have restricted the generalizability of the findings. Additionally, the study focused on molecular identification and genetic diversity rather than developmental biology. Growth curves, which describe the developmental rates of insects under varying temperature conditions, are essential for estimating the postmortem interval (PMI) with precision. However, to the best of our knowledge, there is a lack of comprehensive thermobiological studies for blow fly species in Pakistan. This gap highlights the need for future research to establish baseline developmental data for species such as *C. megacephala*, *C. saffranea*, and *C. rufifacies* under controlled laboratory conditions. This data would complement the molecular database and significantly enhance the identification of the three fly species used in forensic investigations.

## 5. Conclusions

This study represents a pioneering effort in forensic entomology in Pakistan, employing DNA barcoding to identify blow fly species collected from cadavers in Lahore. The high fixation index (FST = 0.83992) and distinct phylogenetic relationships revealed by the NJ tree highlight the unique genetic structure of blow fly populations in the region. While this study provides valuable insights into the genetic structure of blow fly populations in Lahore, the low genetic diversity within the populations underscores the need for broader sampling and the use of additional genetic markers to enhance the resolution of forensic analyses. These findings underscore the need for further research to establish a comprehensive database of forensically relevant insects in Pakistan.

## Figures and Tables

**Figure 1 insects-16-00381-f001:**
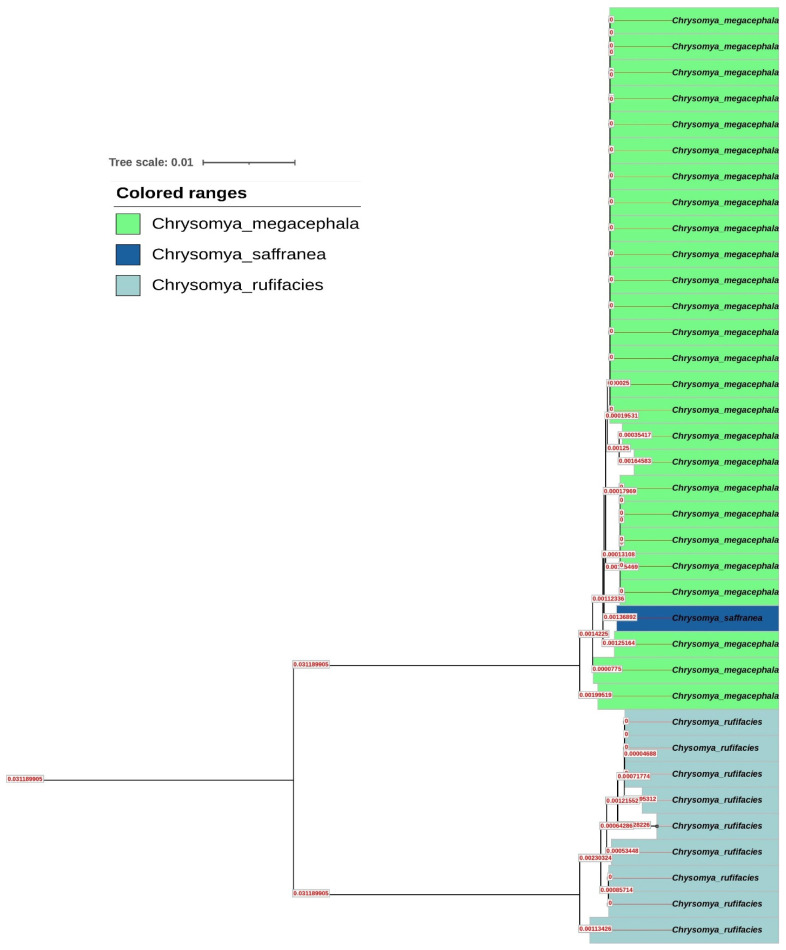
Neighbor-Joining (NJ) tree of sample species.

**Table 2 insects-16-00381-t002:** AMOVA results based on different population groups.

Scheme	d.f.	Sum of Squares	Variance Components	Percentage of Variation
**Among populations**	5	936.066	5.00839 Va	83.99
**Within populations**	294	280.631	0.95453 Vb	16.01
**Total**	**299**	**1216.697**	**5.96292**	**100**
**Fixation Index (FST)**	**-**	**-**	**0.83992**	**-**
**Va and FST *p* (rand. value > obs. value)**	**-**	**-**	**0**	**-**
***p* (rand. value = obs. value)**	**-**	**-**	**0**	**-**
***p*-value**	**-**	**-**	**0.00000 ± 0.00000**	**-**

## Data Availability

The data presented in this study are available on request from the corresponding author, as the raw sequencing files are large and cannot be easily hosted on public repositories.

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
