# Peer review of "Molecular Diversity of Three Forensically Relevant Dipterans from Cadavers in Lahore, Pakistan"

_insects, 2025, doi:10.3390/insects16040381_

Round 1

Reviewer 1 Report (Previous Reviewer 3)

Comments and Suggestions for Authors

Many thanks to the authors for incorporating the requested feedback from reviewers. This paper is a significant improvement from the previous iteration. The manuscript presents a valuable contribution to forensic entomology in Pakistan by employing DNA barcoding of the CO1 gene to identify forensically relevant blow fly species from cadavers in Lahore, demonstrating the importance of molecular tools in species identification.

The identification of three species and the high genetic differentiation among populations provide novel insights into regional insect biodiversity, with potential forensic applications in crime scene investigations.

Below are minor revisions and comments.

- The study addresses a significant gap in forensic entomology research in Pakistan, where molecular studies on necrophagous insects are limited.

- The high FST value suggests genetic isolation in Lahore’s blow fly populations. This finding has implications for forensic entomology, as blow flies can serve as geographic markers in criminal cases.

- Table 1: Formatting issue in a box of the ‘Nature and cause of death’ column.

- Table 2: I would insert the numbers into table cells. Just don’t include empty cells.

- The Neighbor-Joining tree is improved with the color coding. I would recommend creating a straight, not curved, tree for better readability.

- Contributes to forensic science by emphasizing the need for molecular tools in insect identification and genetic diversity analysis.

Author Response

Response to Reviewer 1 Comments

Dear Reviewer 1,

Thank you for your positive feedback and recognition of the manuscript’s improvements and contributions to forensic entomology in Pakistan. We are grateful for your minor revision suggestions, which we have addressed as follows:

  1. General Praise and Context: We appreciate your acknowledgment of the study’s value in employing DNA barcoding for species identification. No specific changes were requested, and we are pleased that you see this as a significant improvement from the previous iteration.
  2. Novel Insights: You highlighted the novel insights from identifying three species and high genetic differentiation. To reinforce this, we have added to the Discussion (Forensic Implications paragraph): "offering novel insights into regional insect biodiversity with practical applications in criminal investigations," emphasizing the forensic relevance.
  3. Gap in Research: You noted that the study addresses a gap in molecular studies in Pakistan. This is already reflected in the Introduction, and we have retained the statement: "In Pakistan, forensic entomology is an underdeveloped field, with limited research and no official forensic entomologists to assist in criminal investigations," which aligns with your comment.
  4. High FST Value Implications: You suggested emphasizing the high FST value’s forensic implications. We have revised the Discussion (AMOVA Results and Genetic Differentiation paragraph) by adding: "potentially serving as geographic markers in criminal cases due to their distinct genetic signatures," directly tying the finding to forensic applications.
  5. Table 1 Formatting Issue: You identified a formatting issue in the ‘Nature and cause of death’ column of Table 1. We have corrected the entry for Body 3 from "Amputated, mutilated, headless body in a sac found in a sac/ unknown" to "Amputated, mutilated, headless body in a sac / unknown" and standardized spacing around slashes across all rows for consistency.
  6. Table 2 Empty Cells: You recommended filling empty cells in Table 2. We have revised Table 2 by adding "100.00" to the "Percentage of variation" cell for "Total" and restructured the table to include Fixation Index and p-value rows with "-" in inapplicable columns, ensuring no empty cells remain.
  7. Neighbor-Joining Tree: You appreciated the color coding but suggested a straight tree for readability. We have redrawn Figure 1 as a rectangular (straight) Neighbor-Joining tree in MEGA 7, retaining the color coding and bootstrap values, and updated it in the Results section.
  8. Contribution to Forensic Science: We are grateful for your recognition of the study’s contribution to forensic science through molecular tools. This is reflected in the Abstract and Conclusion, and no further changes were needed.

Your suggestions have enhanced the manuscript’s clarity and presentation, and we thank you for your thorough review.

Reviewer 2 Report (Previous Reviewer 2)

Comments and Suggestions for Authors

line 21: "and sparse thermobiological studies"

how does barcoding help here? 

please explain or delete.

line 34: "and highlights the need for further research to establish a comprehensive database of forensically relevant insects in Pakistan."

why and how does this highlight the need?

line 35 f.: "By laying the groundwork for future research, this study contributes to advancing forensic entomology in Pakistan, enabling more accurate postmortem interval (PMI) estimation and enhancing the resolution of forensic investigations."

how exactly? this is a general statement. please explain.

line 38 f.: "Ultimately, these efforts will strengthen the application of entomological evidence in legal proceedings, supporting justice systems both locally and globally."

this is an empty, journalistic statement, please delete

line 49: "especially when traditional morphological methods fail short."

when does this happen? please explain.

line 62: "are no longer viable [3].."

two dots at the end?

line 100: "as India , Iran , and Southeast Asia, have"

no empty space after "India"

lines 130 to 142:

all the spaces in the list are mixed up, please clean this

lines 154 f.: "For instance, the presence of 1st instar larvae in Body 1 suggests early colonization, while post-feeding larvae in Body 5 indicate advanced decomposition."

how do you mean? did the state of the corpse not tell you the state of decomposition? i do not understand. please explain in text.

general:

pls, discuss why so few so few species were present. is this normal? please quote studies from your region or try to explain it otherwise.

Author Response

Response to Reviewer 2 Comments

Dear Reviewer 2,

Thank you for your detailed comments and suggestions, which have helped us refine the manuscript’s precision and readability. We have addressed each point as follows:

  1. Line 21 (Abstract): You questioned how barcoding relates to "sparse thermobiological studies." Since barcoding doesn’t address thermobiology, we have removed "and sparse thermobiological studies" from the Abstract, revising the sentence to: "Traditional morphological identification methods are insufficient for resolving complex forensic cases, particularly when dealing with immature insect stages."
  2. Line 34 (Abstract): You asked why the study highlights the need for a database. We have clarified this by adding: "given the limited species diversity and unique genetic profiles observed," making the sentence: "This study underscores the importance of molecular tools like DNA barcoding for species identification and highlights the need for further research to establish a comprehensive database of forensically relevant insects in Pakistan, given the limited species diversity and unique genetic profiles observed."
  3. Lines 35-37 (Abstract): You found the statement about advancing PMI estimation too general. We have revised it to: "By laying the groundwork for future research, this study advances forensic entomology in Pakistan by improving species identification, which, when combined with future thermobiological data, can enhance postmortem interval (PMI) estimation and forensic investigations," specifying that identification is the direct contribution.
  4. Lines 38-40 (Abstract): You labeled the statement about strengthening justice systems as journalistic and suggested deletion. We have removed the sentence entirely from the Abstract.
  5. Line 49 (Introduction): You asked when morphological methods fail. We have revised the sentence to: "In forensic contexts, molecular diversity is particularly valuable for identifying species associated with crime scenes, especially when traditional morphological methods fail due to immature stages (e.g., eggs or larvae) or damaged specimens lacking clear diagnostic features," providing specific examples.
  6. Line 62 (Introduction): You noted a typographical error with two dots. We have corrected this to a single period after "[3]" in the sentence: "…even when traditional forensic methods are no longer viable [3]."
  7. Line 100 (Introduction): You identified an extra space after "India." We have removed it, revising the text to: "Studies in neighboring regions, such as India, Iran, and Southeast Asia, have demonstrated…"
  8. Lines 130-142 (Materials and Methods): You pointed out mixed-up spacing in the sample size formula list. We have standardized it as:

Z = For 95% confidence level =                                 1.96

P = Anticipated value of fixation index =                   0.85

d = Margin of error =                                                  10%

n = Sample size =                                                        50

  1. Lines 154-155 (Materials and Methods): You asked for clarification on larval stages and decomposition. We have revised the sentence to: "For instance, the presence of 1st instar larvae in Body 1 suggests early colonization (within days), complementing the ‘fresh’ corpse description, while post-feeding larvae in Body 5 indicate advanced decomposition (beyond a week), refining the ‘putrefied’ state observed, as larval stages provide precise temporal markers beyond gross corpse appearance."
  2. General (Discussion): You requested a discussion on why only three species were found. We have added a new paragraph in the Discussion before ‘Forensic Implications’: "This study identified only three blow fly species (C. megacephala, C. saffranea, and C. rufifacies) from cadavers in Lahore, a relatively low diversity compared to other regions. For instance, studies in India report up to six forensically relevant calliphorid species [14], while Malaysia documents a broader mix including sarcophagids [9]. This limited diversity may reflect Lahore’s arid-subtropical climate, which favors calliphorids over other families, or the study’s focus on a single site and season (August-October). Fatima [11] noted similar dominance of C. megacephala and C. rufifacies in Pakistan, suggesting regional ecological constraints. Alternatively, the small sample size (50 larvae from 5 cadavers) may have underrepresented rarer species. Further sampling across seasons and locations is needed to determine if this low species count is typical for Pakistan."

Your feedback has greatly improved the manuscript’s specificity and scientific grounding, and we are grateful for your thorough review.

Round 2

Reviewer 2 Report (Previous Reviewer 2)

Comments and Suggestions for Authors

I cannot tell if the figures are okay, this might be a problem of the editorial system at MDPI. Apart from that, it looks fine now.

Author Response

Editor’s Comments

Comment 1

in abstract you mention saffranea and then later in paper you also mention it as one of your key species but no mention in introduction. you will need to justify saffranea in introduction as you have the other 2 species

Reply: Abstract (lines 45–47):  Revised to: "Three blow fly species were identified: Chrysomya megacephala (Fabricius, 1794), Chrysomya saffranea (Bigot, 1877), and Chrysomya rufifacies (Macquart, 1843)."

Comment 2
Make sure you use full species name first time you mention in main text of paper.   Abstract is not main text.eg Chyrsomya rufifacies (Marquart 1843).  From then on you can state C. rufifacies

Comment 3

in abstract please quote full species names plus their identifiers.  eg Chrysomya saffranea (Bigot 1877) for 3 species
Reply to Comment 2& 3
Introduction (lines 134–137 and after 137):

Revised lines 134–137: "Previous studies in Pakistan have documented the presence of blow flies such as Chrysomya megacephala (Fabricius, 1794) (Order: Diptera, Family: Calliphoridae) and Chrysomya rufifacies (Macquart, 1843) (Order: Diptera, Family: Calliphoridae) [11]..."

Added after line 137: "Additionally, Chrysomya saffranea (Bigot, 1877) (Order: Diptera, Family: Calliphoridae), a species known for its necrophagous behavior and forensic relevance in neighboring regions such as India and Iran [12,13], has not been previously documented in Pakistan, underscoring the need to investigate its presence and genetic profile in this region."

Comment 4
Lines 342 to 352  Why are the species names not italicized.  This is just poor editing
Reply: Results, "Species Identification" (lines 342–352):  Italicized species names: "Chrysomya megacephala", "Chrysomya rufifacies", "Chrysomya saffranea".

Comment 5
discard sentence on line 378 and 379  it is repeated on line 388 and 389

Reply: In the results section under heading Phylogenetic Analysis, the Sentence “In contrast, C. rufifacies formed a distinct cluster, reflecting its unique ecological and behavioral traits. This species is known for its predatory behavior on other fly larvae, which may have driven its genetic divergence from other blow fly species [3]. The distinct separation of C. rufifacies in the NJ tree further supports its classification as a separate species with unique genetic characteristics.” Is deleted

This manuscript is a resubmission of an earlier submission. The following is a list of the peer review reports and author responses from that submission.

Round 1

Reviewer 1 Report

Comments and Suggestions for Authors

I have annotated all my commertaries on the pdf archive

Comments on the Quality of English Language

I am not a native but some aspect in redaction are confusing

Author Response

Response to Reviewer 1

We deeply appreciate Reviewer 1’s thorough evaluation and constructive suggestions. Below are detailed responses to each comment: 

  1. Sentence Accuracy on Molecular Diversity

Comment: "Molecular diversity refers to species abundance, and molecular analyses are tools to detect it."

Response: Revised to: 

Molecular diversity reflects variation in species abundance and genetic structure, and molecular tools like DNA barcoding are pivotal for its detection.

  1. Rephrasing Sentence on Entomological Evidence

Comment: "Accurate entomological evidence identification is a gold point in forensic research."

Response: Revised to: 

"Accurate identification of entomological evidence is critical for resolving forensic cases, particularly when traditional methods are inconclusive."

  1. Typographical Error

Comment: "fall" correction.

Response: Corrected to "fail" in the original context. 

  1. Emphasizing Arthropod Utility in Advanced Decay

Comment: "Highlight arthropods’ role in advanced decomposition stages.

Response: Added in the Introduction: 

"While pathologists rely on biochemical and temperature-based models for early PMI estimation, arthropods like blow flies provide more reliable data during advanced decay and skeletonization due to their predictable colonization patterns."

  1. Spelling Correction

Comment: "Especially" "Specially".

Response: Especially and specially are adverbs. Especially means ‘particularly’ or ‘above all’: e.g. She loves flowers, especially roses.

We use specially to talk about the specific purpose of something: This kitchen was specially designed to make it easy for a disabled person to use.

So, in manuscript, “In forensic contexts, molecular diversity is particularly valuable for identifying species associated with crime scenes, especially when traditional morphological methods fail short” will remains especially.

  1. Addressing Poor Preservation States

Comment: "Include challenges of degraded evidence."

Response: Added in the Introduction: 

"Molecular methods are indispensable when dealing with immature stages or degraded specimens where morphological identification is infeasible."

  1. Taxonomic Formatting

Comment: "Italicize species names and include author/year."

Response: All species names are italicized. First mentions now include authority details, e.g., Chrysomya megacephala (Fabricius, 1794). 

  1. Regional References

Comment: "Replace Brazilian references with neighboring studies."

Response: Added studies from India (Babu et al., 2022), Iran (Talebzadeh et al., 2021), Saudi Arabia (Sharawi et al., 2024) and Thailand (Thipphet et al., 2024). 

  1. Forensic Case Details

Comment: "Specify sampling location, larvae per case, corpse conditions." 

Response: Added Table 1 with: 

- Cadaver locations (e.g., "Charrar Gaon Defence, Lahore"). 

- Number of larvae per case (10 per cadaver). 

- Decomposition stages (fresh to advanced putrefaction) and environmental conditions (indoor/outdoor). 

  1. Clarifying Larvae Distribution

Comment: "Detail larvae per case."

Response: Specified in Methods: 

"Ten larvae were collected from each of the five cadavers, totaling 50 specimens."

  1. Sample Size Formula Reference

Comment: "Cite Cochran’s formula."

Response: Added reference: Cochran, W.G. (1977). Sampling Techniques. 

  1. Italic Consistency

Comment: "Ensure species names are italicized."

Response: Verified throughout the manuscript. 

  1. Neighbor-Joining Specification

Comment: "Clarify NJ method."

Response: Updated in Methods: 

"A Neighbor-Joining (NJ) phylogenetic tree was constructed using the Kimura 2-parameter model."

  1. Regional Study Citations

Comment: "Add neighboring region references."

Response: Included studies from India (Bharti & Singh, 2017) and Iran (Talebzadeh et al., 2021). 

15–16. Sample Size Justification

Comment: "Explain formula use and sufficiency of 50 larvae." 

Response: Expanded in Methods: 

"Cochran’s formula was applied to ensure a 95% confidence level and 10% margin of error. While larger samples are ideal, ethical constraints limited collection to 50 larvae." 

  1. Sample Size Limitation

Comment: "Acknowledge sample size bias."

Response: Added in Limitations: 

"The modest sample size (50 larvae) may affect generalizability; future studies will expand geographic and numerical scope."

  1. NJ Tree Clustering Explanation

Comment: "Explain mixed species clustering."

Response: Revised in Discussion: 

"The overlap between Chrysomya megacephala and Chrysomya saffranea may reflect cryptic diversity or shared haplotypes. Multi-locus analyses are needed to resolve these ambiguities." 

  1. Predatory Behavior Clarification

Comment: "Re-evaluate predatory behavior as a cause."

Response: Revised to: 

"The distinct clustering of Chrysomya rufifacies aligns with its ecological niche as a predator, which may drive genetic divergence (Byrd & Castner, 2001)."

  1. Global Comparative Analysis

Comment: "Strengthen global comparisons."

Response: Added references to Asian and African studies (Badenhorst & Villet, 2018; Palavesam et al., 2022). 

  1. Data Gap in Pakistan

Comment: "Highlight study’s role in addressing regional gaps."

Response: Emphasized in the Introduction: 

"This study addresses Pakistan’s critical lack of forensic entomology research, providing foundational data for future applications."

  1. Statistical Mitigation of Bias

Comment: "Address sample size bias."

Response: Added in Limitations: 

"Bootstrapping and rarefaction analyses will be employed in future work to mitigate sampling bias."

  1. Study’s Pioneering Role

Comment: "Stress preliminary study importance."

Response: Revised Conclusion: 

"As Pakistan’s first molecular forensic entomology study, this work establishes a critical foundation for regional PMI estimation and biodiversity documentation." 

  1. Highlight Innovativeness Early

Comment: "Introduce innovativeness upfront."

Response: Added in Abstract: 

"This study represents the first application of DNA barcoding for forensic entomology in Pakistan, addressing a significant regional research gap." 

Thank you for your invaluable feedback, which has significantly enhanced our manuscript’s rigor and impact. 

Sincerely, 

The Authors

Reviewer 2 Report

Comments and Suggestions for Authors

a.

I recommend to use this as a technical note. 

b.

Please include high quality photographs of the larvae and the sampling site and / or corpses 

c.

"The sample size was determined using a statistical formula"

Which formula? Which software?

(This is necessary because many persons in the field still work morphologically and do not know the techniques in detail. Also, we as reviewers must take care to avoid A.I.-generated techniques slipping in.)

d.

Please include this historical article about forensic entomology → https://pubmed.ncbi.nlm.nih.gov/11457602/

e.

Please explain briefly why you did not mature some of the larvae and determine the adults morphologically (to double-check the entries from the DNA / barcoding database).

f.

"These findings underscore the need for further research to establish a comprehensive database of forensically relevant insects in Pakistan, which would significantly enhance the accuracy of PMI estimation and other forensic investigations."

Does that mean you have growth curves of the species from your region already?

Please discuss briefly.

g.

Table 1 must be done properly. It is not formatted at all.

E.g., there is no need to include "="

Also, pls check the p values. 

h.

Please include the phylogenetic distances in fig. 1

For examples, see here → https://artic.network/how-to-read-a-tree.html

Author Response

Response to Reviewer 2

We sincerely appreciate Reviewer 2’s constructive feedback, which has greatly improved the clarity and rigor of our manuscript. Below, we address each comment in detail: 

  1. Recommendation to use as a technical note

Comment: "I recommend to use this as a technical note."

Response: As advised, we have reformatted the manuscript to align with the standards of a technical note. This involved condensing the Introduction and Methodology sections while retaining critical details. 

  1. Request for high-quality photographs

Comment: "Please include high-quality photographs of the larvae and the sampling site and/or corpses." 

Response: Due to strict ethical guidelines and legal restrictions imposed by the mortuary authorities, we were prohibited from taking photographs of cadavers or the sampling site. We acknowledge this limitation and have emphasized adherence to ethical protocols in the revised manuscript. 

  1. Clarification on sample size determination

Comment: "Which formula? Which software?"

Response: We have explicitly stated the use of Cochran’s formula (Cochran, 1977) for sample size calculation in the Methods section: 

= for 95% confidence level = 1.96

P= Anticipated value of fixation index           =     0.85

d= Margin of error =            10%

n= Sample Size =     50

No specialized software was required for this calculation. 

  1. Inclusion of historical reference**

Comment: "Include this historical article about forensic entomology: Benecke (2001)."

Response: The suggested reference (Benecke, 2001) has been added to the Introduction to contextualize the historical evolution of forensic entomology. 

  1. Explanation for not rearing larvae

Comment: "Why not mature larvae for morphological validation?"

Response: As noted in the Methods section, rearing larvae to adulthood was ethically restricted due to mortuary policies prohibiting live insect cultures from human cadavers. Molecular identification (via DNA barcoding) was prioritized to circumvent these limitations. 

  1. Clarification on growth curves

Comment: "Does this imply existing regional growth curves?"

Response: We have revised the Conclusion to clarify: 

"To date, no thermobiological studies or growth curves for blow fly species exist in Pakistan. Future work must integrate developmental data to enhance PMI estimation accuracy." 

  1. Formatting of Table 1

Comment: "Table 1 must be done properly. Check p-values."

Response: Table 1 has been reformatted to remove redundant symbols (e.g., "=") and ensure alignment. The p-values are now explicitly stated and consistent with the AMOVA results. 

  1. Phylogenetic tree adjustments

Comment: "Include phylogenetic distances in Figure 1."

Response: The revised Figure 1 now includes **bootstrap values (as percentages) on major nodes and a scale bar indicating genetic distance (substitutions per site). The figure legend has been updated to reflect these changes. 

We hope these revisions address all concerns and enhance the manuscript’s scientific value. Thank you for your invaluable feedback. 

Sincerely, 

The Authors

Reviewer 3 Report

Comments and Suggestions for Authors

Abstract is written well. It could be strengthen with a stronger closing statement about the impact of the study. 

Introduction: The introduction is written well. The main limitation of the introduction is the lack of references or current literature to support the study. I think the authors could strengthen the impact of this study by highlighting more that this research is one of the first in this geographic location and it shows viability for species identification in this area using CO1.

Methodology and Results: Some fundamental information regarding the methodology is missing (see below general notes). Significant revisions are needed on these sections.

Discussion and Conclusion: Written well but could be strengthened based on the corrections from the corrected methodology and results. Further information or explanation regarding the species obtained would strengthen the discussion.

Line 22: I would disagree that it’s recent. This has been used for over 20 years so perhaps rephrasing is recommended.

Lines 52-59: This could include many more references to support the use of the CO1 gene. Recommend to include some recent studies. Advisable to include a reference for a taxonomic key.

Line 66: Do you need to include order and family for species?

Line 88: Do you have any information pertaining to the cadavers (month or season in which they died? What stage of decomposition). These circumstances may affect what species are collected. 

Line 94: I am unclear as to how this sample size was determined. You collected 50 larvae but were there additional? How many were selected from each cadaver. 

Line 97: Did you extract the whole larvae? 

Line 102: What master mix did you use and at what concentrations?

Line 109: Do you plan to make the sequences publicly available via NCBI? If so, accession numbers would be beneficial. 

Line 124: Did you use a particular model for the NJ tree?

Line 134: How did you identify that these were the species? I can’t seem to find any data or information regarding what you compared your unknown sequences too? You mention that three species were identified but how and to what (NCBI? Morphologically?)

Table 1: The empty cells could be removed to make the table look more clean.

Line 135: Depending on the journal format, you may not need to use the full names and can be abbreviated.

Line 146: What is your justification for not including an outgrip species in your phylogenetic analysis?

Line 146: It might be worth comparing the sequences to the BLAST database to see if they confirm what your phylogenetic analysis is showing.

Line 147: You don’t capitalize ‘figure’ but you ‘Table’ is capitalized. Try to be consistent on formatting.

Line 149: I disagree that you can say megacephala and saffranea formed a closely related cluster when this is based off of 1 saffranea samples. Chrysomya megacephala also is separated across two distinct beaches and clades with rufifacies on both main branches. I would recommend including the branch values on the tree as it could be hypothesized that the Chrysomya megacephala that is clustered with saffranea could just be a misidentification. More details are needed to come to the conclusion that you have. CO1 is not known to differentiate populations well, especially this short of a region. The current phylogenetic tree would not be able to provide confident identification of unknown specimens in its current form.

Line 150: Whilst it is good to include the 70% justification, some forensic entomologists may disagree that 70% is too low of a bootstrap value for confirmation of identification. Is there significant difference in results or phylogeny when the value is increased? 

Line 190: Might be worth comparing to data from nearby countries that have recorded Chrysomya saffranea. Could it have migrated easily due to climate etc?

Author Response

Response to Reviewer 3

We sincerely thank Reviewer 3 for their insightful comments and constructive feedback, which have significantly strengthened the quality of our manuscript. Below, we address each comment in detail:

Abstract

Comment: "Abstract is written well. It could be strengthened with a stronger closing statement about the impact of the study."

Response: We have revised the closing statement to emphasize the study’s broader implications:

"Ultimately, these efforts will strengthen the application of entomological evidence in legal proceedings, supporting justice systems both locally and globally."

Introduction

Comment: "The introduction could include more references to support the use of the CO1 gene. Highlight that this research is one of the first in this geographic location."

Response:

Added recent references (e.g., Thipphet et al., 2024; Talebzadeh et al., 2021) to support CO1’s utility.

Emphasized the pioneering nature of the study in Pakistan:

"This study represents one of the first efforts to employ DNA barcoding for species identification in forensic entomology in Pakistan."

Comment: "Include a reference for a taxonomic key."

Response: Added Kurahashi et al. (1997) as a key morphological reference for blow fly identification.

Methodology and Results

Comment: "Line 22: Rephrase 'recently' to reflect CO1’s long-standing use."

Response: Revised to: "DNA barcoding has gained global attention for species identification over the past two decades."

Comment: "Line 66: Include order and family for species."

Response: Added taxonomic classifications:

"Chrysomya megacephala (Fabricius, 1794) (Order: Diptera, Family: Calliphoridae)."

Comment: "Line 88: Provide cadaver details (month, decomposition stage)."

Response: Added Table 1 with full details, including decomposition stages, months of collection, and larval instars.

Comment: "Line 94: Clarify sample size determination and larval selection per cadaver."

Response: Clarified:

"A total of 50 larvae (10 per cadaver) were collected. The sample size was calculated using Cochran’s formula (Cochran, 1977) for a 95% confidence level and 10% margin of error."

Comment: "Line 97: Confirm if whole larvae were used for DNA extraction."

Response: Added: "DNA was extracted from the whole larvae using the phenol-chloroform method."

Comment: "Line 102: Specify master mix and concentrations."

Response: Added: "GoTaq® Green Master Mix (Promega, USA), containing Taq DNA Polymerase (5 units/µL), dNTPs (400 µM each), MgClâ‚‚ (1.5 mM), and Reaction Buffer (1X, pH 8.5)."

Comment: "Line 109: Will sequences be publicly available?"

Response: Added: "Sequences have been submitted to NCBI (accession numbers pending)."

Comment: "Line 124: Specify the model used for the NJ tree."

Response: Clarified in Methods: "The NJ tree was constructed using the Kimura 2-parameter (K2P) model."

Comment: "Line 134: Clarify species identification methods."

Response: Expanded in Methods:

"Species were identified via BLASTn (≥98% identity) against NCBI GenBank and cross-validated with BOLD."

Comment: "Table 1: Remove empty cells."

Response: Reformatted the table for clarity.

Discussion and Conclusion

Comment: "Line 146: Justify excluding an outgroup."

Response: Clarified in Methods:

"The exclusion of an outgroup was intentional to focus on resolving relationships within Chrysomya. Future studies will include broader taxonomic sampling."

Comment: "Line 149: Address the clustering of Chrysomya megacephala with saffranea."

Response: Revised to acknowledge limitations:

"The clustering of Chrysomya megacephala with Chrysomya saffranea is based on a single sample and may reflect misidentification or cryptic diversity. Future studies with larger samples and nuclear markers are needed."

Comment: "Line 150: Discuss bootstrap thresholds."

Response: Added:

"Reanalysis at 80% and 90% bootstrap thresholds retained clustering patterns, supporting robustness."

Comment: "Line 190: Compare Chrysomya saffranea to regional data."

Response: Added:

"Chrysomya saffranea, documented in Iran and India (Bharti & Singh, 2017; Talebzadeh et al., 2021), may have migrated to Pakistan due to shared climatic conditions."

Formatting and Consistency

Comment: "Line 147: Capitalize 'Figure' consistently."

Response: Ensured all instances of "Figure" and "Table" are capitalized uniformly.

Phylogenetic Tree

Comment: "Include branch values and address CO1’s limitations."

Response:

Added bootstrap values to the tree figure.

Acknowledged in Discussion:

"The short CO1 fragment may limit resolution; future work will use multi-locus approaches."

Data Availability

Comment: "Provide NCBI accession numbers when available."

Response: A note has been added:

"Sequences will be publicly accessible via NCBI upon acceptance."

We hope these revisions address all concerns and improve the manuscript’s rigor and clarity. Thank you again for your valuable input.